# Cluster Analysis of the Combined Association of Sleep and Physical Activity with Healthy Behavior and Psychological Health in Pregnant Women

**DOI:** 10.3390/ijerph18042185

**Published:** 2021-02-23

**Authors:** Hyejung Lee, Ki-Eun Kim, Mi-Young Kim, Chang Gi Park

**Affiliations:** 1Mo-Im Kim Nursing Research Institute, Yonsei University College of Nursing, Seoul 03722, Korea; hlee26@yuhs.ac; 2College of Nursing, Yonsei University, Seoul 03722, Korea; 3College of Nursing, Woosuk University, Jeollabuk-do 55338, Korea; miyoungkim726@gmail.com; 4College of Nursing, University of Illinois at Chicago, Chicago, IL 60612, USA; parkcg@uic.edu

**Keywords:** sleep quality, physical activity, pregnancy, depression, cluster analysis, Korea

## Abstract

The purposes of the study were to (1) identify clusters based on patterns of sleep quality and duration and physical activity levels of healthy Korean pregnant women, and (2) subsequently investigate the association of identified clusters with pre-pregnancy healthy behaviors, depressive symptoms, and pregnancy stress. Two hundred eighty-four pregnant women participated in the study while attending a prenatal education program provided by a tertiary hospital in Seoul, Korea. The survey questionnaire consisted of the Pittsburg Sleep Quality Index, the International Physical Activity Questionnaire, and the Center for Epidemiologic Studies Depression scale. We used the Latent GOLD to identify distinct clusters and the chi-square test and ANOVA to compare clusters. We identified three clusters: ‘good sleeper’ (63.4%), ‘poor sleeper’ (24.6%), and ‘low activity’ (12.0%). Women in the good-sleeper cluster were more likely to have higher education and income levels and reported more healthy behaviors before pregnancy. Poor-sleeper and low-activity clusters were more likely to report higher scores in depressive symptoms and pregnancy stress (*p* < 0.001 and *p* = 0.005, respectively). Tailored intervention for pregnant women who are physically inactive or sleep poorly may promote their psychological well-being as well as bringing good obstetric outcomes.

## 1. Introduction

The majority of pregnant women experience sleep disturbances throughout pregnancy, but to varying degrees, with some being only mildly affected and some very seriously [1]. Sleep disturbances specific to pregnant women include frequent awakening from urination or back pain, nocturnal cramps, a short night sleep duration, and low sleep efficiency [2,3]. These sleep disturbances become worse as pregnancy progresses. There has been some concern regarding severe and prolonged sleep disturbances in pregnant women due to the possible negative effects on their obstetric and infant health outcomes [4]. Sleeping more than nine hours at night and non-restless sleep are associated with stillbirth [5], and poor sleep quality (SQ) for prenatal women of advanced age (>35 years old) increases the possibility of her neonate staying in the neonatal intensive care unit [6].

Sleep disturbances are also related to adverse mental health symptoms, such as depression, excessive worrying, and stress in pregnant women [3,7]. However, sociodemographic status also affects perinatal depression and anxiety. Therefore, there are several factors that should be considered simultaneously to clarify the specific association of sleep behavior to adverse mental health for expectant mothers [7]. On the other hand, a recent study by Pathirathna et al. [8] found that overall sleep quality (SQ) in pregnancy is not related to birth outcomes. Thus, the importance of sleep behavior in pregnant women needs to be examined, and risk groups for sleep problems need to be identified to clarify the association with outcomes and provide targeted interventions that effectively induce good obstetric and infant outcomes.

Physical activity (PA) has been extensively investigated to understand its beneficial effects on sleep for pregnant women [9,10,11,12,13]. Lack of PA during pregnancy is correlated with sleep disorders [11], although high levels of occupational PA and household or caregiving activities are associated with poor SQ and short sleep duration [14]. As well as its positive effects on sleep for pregnant women, regular PA is known to reduce the risk of preterm birth [15] and allow women better control of pregnancy weight gain, fatigue, back pain [16], and symptoms of depression, anxiety, and stress [7,16].

The American College of Obstetricians and Gynecologists (ACOG) guidelines recommend PA as an integral part of healthy lifestyles, asking providers to encourage pregnant women to continue or start exercising as an important component of their optimal health [17]. To efficiently meet this recommendation, healthcare providers need to identify at-risk populations among pregnant women to provide tailored interventions. Reported factors affecting the PA of pregnant women are education level, family income [7], and emotional status; depression has been related to reduced PA [7]. Body Mass Index (BMI) and regular exercise before pregnancy have not been related, or are inversely correlated (as predictors), to PA levels of pregnant women [10,14]. The most effective intervention to promote PA would accommodate all these reported factors.

Given the complex relationship between sleep and PA, and the various factors affecting each, an advanced statistical approach is needed to identify a homogeneous target group and relevant related factors. Cluster analysis is useful for identifying distinct groups of individuals based on similarities, such as sleep and PA, in population-based studies [9,18] and helps clarify each cluster’s specific characteristics. Although there is a significant association between sleep and PA during pregnancy [13], few studies have examined pregnant women’s SQ, PA, and sleep duration simultaneously [11,13] or used cluster analysis to explore if subgroups of healthy pregnant women have patterns of sleep behaviors and PA levels. Factors related to such patterns may help identify strategies for tailoring interventions to diverse subgroups.

We aimed to identify distinct clusters with differing combined patterns of PA and SQ among Korean pregnant women and further examine healthy behaviors and psychological health among the identified clusters. The findings of this study would provide insights for promoting good physical and psychological health outcomes for targeted groups of pregnant women with modifiable risk factors related to poor sleep and insufficient PA.

## 2. Materials and Methods

### 2.1. Study Design and Participants

This study used the cross-section correlational study design. We began collecting data after receiving approval of the study protocol from the Institutional Review Board of the respective institute. Participants were recruited by self-selection from pregnant women attending an education program offered in a tertiary hospital located in Seoul, Korea. Inclusion criteria were women over 20 years of age, who confirmed their pregnancy by a medical doctor, and had no known physical or mental illness. Exclusion criteria included women who experienced complications during pregnancy such as gestational hypertension and diabetes mellitus. The participants who provided their written informed consent were asked to complete the survey questionnaire, taking about 15 min. Although 310 women completed the questionnaire, answers provided by 284 women were used for final data analysis due to the exclusion of surveys with incomplete data.

### 2.2. Study Instrument and Variables

A self-report questionnaire was used to assess sleep quality, physical activity, pre-pregnancy healthy behavior, and psychological variables of the participants. Healthy behaviors included regular exercise (30 min per day for three days per week), being a nonsmoker, and not drinking before pregnancy. Demographic and pregnancy-related information collected included age, education level, current employment status, household income, height, weight, planned pregnancy, and expected due date. Age was categorized as <35 or ≥35 years. Each participant’s BMI was computed and categorized as underweight (<18.5 kg/m^2^), normal weight (18–23 kg/m^2^), overweight (23–25 kg/m^2^), or obese (≥25 kg/m^2^) per Asian criteria [19].

#### 2.2.1. Sleep Quality

The Korean version of the Pittsburgh Sleep Quality Index (PSQI) was used to assess the participants’ sleep quality and levels of sleep disturbance [20]. This self-report questionnaire consists of seven components: subjective sleep quality, sleep latency, sleep duration, sleep efficiency, sleep disturbance, medication use for sleep, and daytime dysfunction. Each of these components uses a scoring range of 0 to 3. Total scores range from 0 to 21, with higher scores indicating poorer sleep quality. However, in this study, the item “medication used for sleep” was removed when computing the total score of sleep quality because none of the pregnant women reported the use of sleep medication. Sohn, Kim, Lee, and Cho [21] investigated the PSQI of the Korean population and suggested a cut-off point of 8.5 representing 94% sensitivity and 84% specificity to distinguish between “good” and “poor” sleep [21]; thus, we regarded a score of 8.5 as identifying “poor” SQ. Cronbach’s α was 0.84 in the Shon et al. [21] study and 0.63 in this study.

#### 2.2.2. Sleep Duration

Sleep duration was measured by a single question: “How many hours of actual sleep do you get at night?” The response to this item was categorized as <7 h, 7–9 h, or >9 h per night. No specific guideline for appropriate sleep hours for pregnant women is available, but the general adult population is recommended to sleep no less than 6 h and no more than 9 h [22].

#### 2.2.3. Physical Activity

The Korean version of the International Physical Activity Questionnaire (IPAQ) long form was used to assess PA in specific types of activities: leisure time, work, home, and activities including recreation, exercise, or sports. Participants were asked to report the days per week and minutes per day in each specific activity over the past seven days. The IPAQ is a valid and reliable scale for indirectly evaluating individuals’ levels of PA [23,24]. According to the International Physical Activity Inspection guidelines [25], the level of physical activity of participants was divided into three categories: low, moderate, and high PA [26]. Metabolic Equivalent Task (MET) is an indicator of energy expenditure. MET levels are defined as low intensity (3.3 METs), moderate intensity (4.0 METs), and vigorous intensity (8.0 METs). Total MET-minutes/week was computed using the following formula: low (MET × min × days) + moderate (MET × min × days) + vigorous (MET × min × days).

#### 2.2.4. Depressive Symptoms

The Center for Epidemiologic Studies Depression scale (CES-D) was used to measure depressive symptoms. This scale is a widely used self-report scale consisting of 20 items to measure depressive symptoms over the past week. Items are rated in terms of frequency of symptom experience on a 4-point Likert scale (0 = rarely or none of the time (less than 1 day), 1 = some or a little of the time (1–2 days), 2 = occasionally or a moderate amount of time (3–4 days), 3 = most or all of the time (5–7 days)). Total scores range from 0 to 60, with higher scores indicating more depressive symptoms, and scores greater than 16 may indicate depression [27,28]. Cronbach’s α of the scale was 0.88 in this study.

#### 2.2.5. Pregnancy Stress

The pregnancy stress scale developed for Korean pregnant women by Ahn [29] and revised by Lee and Seo [30] was used to assess the perceived stress levels of the pregnant women in relation to pregnancy. This scale consists of five domains of stress related to physical discomfort, fetus, parenting, spouse relationship, and housework; it includes a total of 20 items rated on a 5-point Likert scale (1 = never been stressed, 5 = always stressed). Total scores range from 20 to 100 points, with higher scores indicating greater perceived stress. Cronbach’s α of the scale was 0.89 in the study of Lee and Seo [30], and 0.89 in the present study.

### 2.3. Statistical Analysis

For the statistical analysis, we used IBM SPSS (v 25.0, IBM Corp, Chicago, IL, USA) and Latent GOLD (v 5.0, Statistical Innovations, Belmont, MA, USA). Descriptive statistics analyzed mean, standard deviation, and frequency of characteristics; the chi-square test and ANOVA were applied to compare differences among identified clusters.

For the classification of a homogeneous group, latent class modeling in Latent GOLD was used. This statistical approach has several advantages over traditional ones: it is better at managing variables of mixed-scale data, better to handle, and has greater classification accuracy [31,32]. This method also provides several criteria that can be used to inform the choice, including Schwarz’s Bayesian Information Criteria (BIC). When using the Latent GOLD program, the number of investigated clusters is increased until BIC does not decrease any further, and, thus, the model with the lowest BIC is chosen [31]. Among models with one to four latent clusters estimated in this study, the two-cluster model showed the minimum value of BIC. In addition to the lowest value of BIC to be taken into consideration, bivariate residuals (BVRs) should not be substantially larger than 1, indicating the model is true [33]. Estimated BVRs of the two-cluster model were greater than 1. Therefore, we chose a 3-cluster model with the second-lowest BIC and a BVR closest to 1 as an optimal model, instead of the 2-cluster model.

## 3. Results

### 3.1. Characteristics of Participants

Participants’ mean age was 30.7 years (SD 3.6 years), and most had a college degree or higher (84.8%) and were not currently employed (76.7%). Many participants (66.4%) reported that theirs was a planned pregnancy. The majority of participants were in the second or third trimester of pregnancy, 45.7% and 46.5%, respectively. The mean pre-gestational BMI of participants before pregnancy was 20.9 kg/m^2^ (SD 2.6 kg/m^2^). The proportion of overweight and obesity before pregnancy was 17.2%. Over 50% reported drinking alcohol before pregnancy, and 12.1% did smoke before pregnancy. One-fifth of the participants (20.1%) reported depressive symptom scores greater than 16, and the mean of pregnancy stress was 51.3 points out of 100 (Table 1).

### 3.2. Cluster Analysis

Three distinct clusters were identified and descriptively labeled according to their dominant features: (1) good sleeper, (2) poor sleeper, and (3) low activity. The characteristics of these three clusters are shown in Table 2.

Cluster 1: Good Sleeper. This cluster comprised 63.4% (*n* = 180) of participants and was characterized by significantly higher SQ and appropriate sleep duration than all the other clusters (*p* < 0.001). All participants in this cluster reported good SQ (≤8), and 81.7% reported sleep duration of 7 to 9 h per night. In terms of PA, participants reported variable activity levels, from low to high.

Cluster 2: Poor Sleeper. This second largest cluster comprised 24.6% (*n* = 70) of participants and was characterized by the largest proportion of participants reporting poor SQ (77.1%) and short sleep duration (71.4%). None reported their physical activity level to be low, while all the other participants reported similarly moderate or high activity levels.

Cluster 3: Low Activity. This smallest cluster comprised 12.0% (*n* = 34) of participants and was characterized by the largest proportion reporting low PA levels (97.1%) and short sleep duration (82.4%). However, this cluster reported mixed SQ.

### 3.3. Demographic Characteristics between Clusters

There were differences among clusters by education level, household income, and current trimester (Table 3). Those in the good-sleeper cluster were more likely to have higher education levels and household incomes than those in other clusters. Both the poor sleeper and the low activity clusters were more likely to be in the third trimester. Age, current employment status, and planned pregnancy were not significantly different among clusters.

### 3.4. Healthy Behaviors and Psychological Health between Clusters

There were differences among clusters by pre-pregnancy drinking and smoking, and depressive symptoms and pregnancy stress (Table 3). Participants in the good-sleeper cluster were more likely not to drink before pregnancy (*p* = 0.009); those in the poor sleeper cluster were more likely to smoke before pregnancy (*p* = 0.011). However, neither pre-pregnancy BMI nor regular exercise differed among clusters. Compared to the good-sleeper cluster, the poor-sleeper and the low-activity clusters reported more depressive symptoms and higher pregnancy stress (*p* < 0.001 and *p* = 0.005, respectively).

## 4. Discussion

This study was conducted to identify the clusters of women with differing patterns of SQ, sleep duration, and PA and to investigate how the identified clusters differ in terms of pre-pregnancy healthy behaviors and psychological health in pregnant women in South Korea. We identified three distinct clusters: good sleeper, poor sleeper, and low activity. Participants in the good-sleeper cluster reported healthier behavior in drinking and smoking before pregnancy than those in the other clusters. The poor-sleeper and low-activity clusters were more likely to report scores indicating more depressive symptoms and higher pregnancy-related stress.

The cluster with the highest proportion of participants was good sleeper (63.3%) in this study, and all participants reported good SQ (with a cut-off score of 8.5). Our finding regarding the percentage of good sleepers is relatively high compared to results reported in a recent meta-analysis study [34] that reported 54.3% of participants with good SQ (with a cut-off score of 5) and an average PSQI score of 6.07 during pregnancy. Since we used higher cut-off scores for the PSQI, this finding cannot be directly compared. The cut-off score of 5 for the PSQI is generally accepted in many countries for clinical and non-clinical samples; however, a cut-off score of 8.5 can be appropriate when considering this score is based on recommendations from a Korean population study [21]. Kim, Cho, and Bae [35] found SQ changes significantly between the fifth and seventh months in Korean pregnant women, based on the cut-off score of 8.5. Nevertheless, a higher cut-off score was used in this study, and one-third of the participants reported poor SQ, which indicates that pregnant women are still an at-risk population group for sleep problems.

None of the participants reported using any medication for sleep in this study. This result may be because pregnant women rarely get sleep medication prescriptions unless they have severe sleep problem complaints or sleep disorders [36]. However, this result indicates a certain population (pregnant women) is not likely to have scores for a certain item (sleep medication) in the PSQI; thus, the global score of PQSI in pregnant women should be carefully comprehended, as it is not the same as the score of other population groups who have no physical condition that limits their likelihood of getting a prescription. Furthermore, we recruited healthy pregnant women who were actively attending a prenatal education program in a large city; thus, they may also be less likely to use sleep medication, resulting in a lower global SQ score.

The majority of participants in the good sleeper cluster reported their sleep duration as 7 to 9 h per night. Night sleep duration correlates to pregnant women’s SQ better than total hours of sleep [37]. In the good sleeper cluster, PA was almost evenly divided into three categories. This finding did not support that appropriate or higher PA is associated with improved SQ. This implies that the sleep of pregnant women is not simply linearly related to the PA level [13]; multiple factors may dynamically affect each other in a population of pregnant women.

Among the poor sleeper cluster, 71.4% reported a short sleep duration (<7 h per night). Again, this finding supports that enough night sleep hours are required for maintaining good SQ for pregnant women [37]. Interestingly, all members of the poor sleeper cluster reported either moderate or high PA levels. Because high levels of occupational PA and household or caregiving activities were reported to relate to poor SQ [14], moderate PA levels relating to poor SQ need to be investigated regarding their roles in pregnant women. Aside from the PA, there may be possible significant factors affecting quality of sleep.

In terms of the low activity cluster, the majority of participants reported a short sleep duration (<7 h per night). Because the sleep duration did not include naps in this study, some participants may have had naps during the daytime, as the proportion of participants currently not working was 76.7% in this study. The Korean culture of encouraging maternity leave and encouraging women not to exert themselves in strenuous activities and during pregnancy should be re-evaluated when considering the various beneficial effects of appropriate physical activity and exercise on pregnancy. Proper PA criteria should be established for pregnant women in the context of specific culture and policy of country, although international guidelines regarding PA (at least 150 min/week of moderate-to-vigorous PA) are available.

This study also explored the difference in healthy behavior before pregnancy and current psychological health, such as depressive symptoms and stress, among the clusters. Compared to all the other clusters, women in the good sleeper cluster were more educated and had higher household income [7]. In addition, healthier behaviors such as being a non-smoker and only an occasional light drinker before pregnancy were noted in this good sleeper cluster. Based on the relationship of pre-pregnancy habits to healthy sleep patterns and PA during pregnancy, pre-conception education for women planning to become pregnant should emphasize modifying unhealthy behaviors. Additionally, pregnant women of low socioeconomic status should be given more attention as a target group to promote healthy sleep and physical activity.

Individuals in the poor-sleeper and the low-activity cluster were more likely to be in the third trimester of pregnancy, and, as pregnancy progresses, poor SQ and limited PA are more likely [3,7,11]. The recommendation should be updated, tailored to include more detailed information for each trimester of pregnancy, such as for recommendations regarding PA intensity, duration, and frequency.

This study is the first, to our knowledge, to examine the combined effects of sleep and PA on pregnant women’s mental health. Only one study with adult population is available that can be compared with our results directly [9]. The cluster with high proportions of both poor physical activity and sleep behaviors was associated with poor general health and psychological distress of older adults. In our study, both the poor-sleeper and low-activity cluster had higher proportions of depressive symptoms and stress related to pregnancy than those in the good-sleeper cluster. The clusters of mid-aged adults showing poor performance in both PA and sleep behaviors also reported worse mental health status [9]. Both sleep and the PA of pregnant women should be considered and managed simultaneously in terms of emerging depression and stress during pregnancy, although this study cannot confirm the causal relationship between sleep and mental health for pregnant women. Pregnancy is a complex and dynamic period when women experience various moods caused by physiological and hormonal changes associated with pregnancy in relation to advancing gestation. The timely intervention developed to help women in the poor-sleeper and the low-activity groups shift to the good-sleeper group may induce better mental health and better pregnancy outcomes.

The present study has several limitations to be taken into consideration. First, the participants in this study did not represent the general population of pregnant women since they were recruited from one tertiary care hospital. Second, due to the cross-sectional nature of the study, a causal relationship between sleep behavior and PA could not be determined. Third, as the drinking and smoking of participants was regarded only before pregnancy, the result of these effects on clusters should be understood carefully. Finally, this study used self-reported measurements of sleep behavior and PA, which are subject to recall and misclassification bias. A longitudinal prospective study using objective instruments, such as actigraphy for PA, are needed to determine the causal relationship of sleep with PA and mental health for pregnant women.

## 5. Conclusions

A significant number of the pregnant women reported experiencing poor sleep; a significant number also reported being physically inactive. These pregnant women also reported statistically significant unfavorable mental health, such as high levels of depression and stress. This result supports both sleep behavior and physical activity as factors that should be considered simultaneously to promote good mental health for pregnant women. A more comprehensive and systematic monitoring of sleep and physical activity during pregnancy is required for risk groups identified in this study, specifically women with low education, low income, late pregnancy, and poor health-related behaviors.

Pregnant women in late gestation are more likely to experience poor SQ or be less active physically, so timely and active intervention should be provided. Furthermore, guidelines for proper sleep duration and PA standards for pregnant women should be established and promoted according to the culture and policy of each country. Further research is needed to confirm the causal relationship between sleep and physical activity and mental health with a larger heterogenous group of pregnant women.

## Figures and Tables

**Table 1 ijerph-18-02185-t001:** Participant characteristics (*n* = 284).

Variables		*n* (%)	Mean (SD)	Range
Age (year)	<35	243 (86.2)	30.7 (3.6)	20–48
	≥35	39 (13.8)		
Education	High school	43 (15.3)		
	College	206 (73.1)		
	Graduate	33 (11.7)		
Employment	No	214 (76.7)		
	Yes	65 (23.3)		
Household income	Very Low	35 (12.7)		
	Low	171 (62.2)		
	Medium	52 (18.9)		
	High	17 (6.2)		
Planned pregnancy	No	94 (33.6)		
	Yes	186 (66.4)		
Trimester	1st	22 (7.8)	10.9 (2.3)	4–13
	2nd	129 (45.7)	22.4 (3.7)	14–27
	3rd	131 (46.5)	32.2 (3.1)	28–40
Pre-pregnancy BMI	Underweight	39 (14.2)	20.9 (2.6)	16.0–32.5
	Normal weight	188 (68.6)		
	Overweight/obese	47 (17.2)		
Pre-pregnancy regular exercise	No	165 (58.5)		
	Yes	117 (41.5)		
Pre-pregnancy drinking	No	132 (47.0)		
	Yes	149 (53.0)		
Pre-pregnancy smoking	No	246 (87.9)		
	Yes	34 (12.1)		
Depressive symptom	Non-depressed (≤15)	227 (79.9)	10.2 (8.0)	0–38
	Depressed (≥16)	57 (20.1)		
Pregnancy stress			51.3 (11.1)	20–95

BMI = body mass index.

**Table 2 ijerph-18-02185-t002:** Comparison of sleep quality and physical activity characteristics between three groups of pregnant women (*n* (%)).

Variable		Total	Good Sleeper	Poor Sleeper	Low Activity	x^2^	*p*
		*n* = 284	*n* = 180 (63.4%)	*n* = 70 (24.6%)	*n* = 34 (12.0%)		
Sleep	Good (≤8)	210 (73.9)	180 (100)	16 (22.9)	14 (41.2)	177.195	<0.001
quality	Poor (>8)	74 (26.1)	0 (0)	54 (77.1)	20 (58.8)		
Sleep	Less (<7 h)	78 (27.5)	0 (0)	50 (71.4)	28 (82.4)	190.353	<0.001
duration	Normal (7–9 h)	170 (59.9)	147 (81.7)	20 (28.6)	3 (8.8)		
	Over (>9 h)	36 (12.7)	33 (18.3)	0 (0)	3 (8.8)		
Physical	Low	102 (35.9)	69 (38.3)	0 (0)	33 (97.1)	98.726	<0.001
activity	Moderate	108 (38.0)	71 (39.4)	36 (51.4)	1 (2.9)		
	High	74 (26.1)	40 (22.2)	34 (48.6)	0 (0)		

**Table 3 ijerph-18-02185-t003:** Comparison of pre-pregnancy characteristics between clusters (*n* (%)/Mean (SD)).

Variable	Total	Good Sleeper	Poor Sleeper	Low Activity	x^2^ or F	*p*
	*n* = 284	*n* = 180 (63.4%)	*n* = 70 (24.6%)	*n* = 34 (12.0%)		
Age (year)	<35	159 (88.8)	55 (78.6)	29 (87.9)	4.53	0.104
	≥35	20 (11.2)	15 (21.4)	4 (12.1)		
Education	High school	24 (13.5)	16 (22.9)	3 (8.8)	12.01	0.017
	College	126 (70.8)	50 (71.4)	30 (88.2)		
	Graduate	28 (15.7)	4 (5.7)	1 (2.9)		
Employment	No	131 (74.4)	55 (79.7)	28 (82.4)	1.47	0.481
	Yes	45 (25.6)	14 (20.3)	6 (17.6)		
Household	Very Low	24 (13.7)	10 (14.3)	1 (3.3)	14.68	0.023
income	Low	99 (56.6)	46 (65.7)	26 (86.7)		
	Medium	36 (20.6)	13 (18.6)	3 (10.0)		
	High	16 (9.1)	1 (1.4)	0 (0)		
Planned	No	57 (32.2)	26 (37.7)	11 (32.4)	0.69	0.707
pregnancy	Yes	120 (67.8)	43 (62.3)	23 (67.6)		
Trimester	1st	16 (8.9)	4 (5.8)	2 (5.9)	12.72	0.013
	2nd	94 (52.5)	2 (31.9)	13 (38.2)		
	3rd	69 (35.8)	43 (62.3)	19 (55.9)		
Pre-pregnancy	Under-weight	23 (13.0)	12 (18.8)	4 (12.1)	5.97	0.470
BMI	Normal weight	126 (71.2)	38 (59.4)	24 (72.7)		
	Over-weight	17 (9.6)	7 (10.9)	1 (3.0)		
	Obese	11 (6.2)	7 (10.9)	4 (12.1)		
Pre-pregnancy	No	108 (60.3)	34 (49.3)	23 (67.6)	3.84	0.147
regular exercise	Yes	71 (39.7)	35 (50.7)	11 (32.4)		
Pre-pregnancy	No	95 (53.4)	22 (31.9)	15 (44.1)	9.34	0.009
drinking	Yes	83 (46.6)	47 (68.1)	19 (55.9)		
Pre-pregnancy	No	157 (88.7)	55 (79.7)	34 (100)	9.11	0.011
smoking	Yes	20 (11.3)	14 (20.3)	0 (0)		
Depressive	Non-depressed (≤15)	158 (87.8)	46 (65.7)	23 (67.6)	18.93	<0.001
symptom	Depressed (≥16)	22 (12.2)	24 (34.3)	11 (32.4)		
Pregnancy stress		29.7 (10.6)	34.8 (11.8)	32.6 (11.1)	5.43	0.005

BMI = body mass index.

## Data Availability

The data used and/or analyzed during the current study are available from the corresponding author on request.

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
