# Peer review of "Cluster Analysis of the Combined Association of Sleep and Physical Activity with Healthy Behavior and Psychological Health in Pregnant Women"

_ijerph, 2021, doi:10.3390/ijerph18042185_

Round 1

Reviewer 1 Report

The purpose of this study was to identify distinct clusters with differing combined patterns of PA and sleep quality among Korean pregnant women and examine healthy behaviors and psychological health among the identified clusters.

In this study, all pregnant women from 1st to 3rd trimesters were included. Sleep and physical activity may differ according to trimester. In this study, the difference between the three groups was not considered in the data analysis.

line 100: Do researchers view smoking and alcohol drinking as healthy behavior?

Line 100-101: It is not appropriate to include weight and height, planned pregnancy, and expected date due in demographic variables.

line 104: I suggest replacing healthy weights with normal weights.

line 108: Need citation for PSQ

line 120: Sleep duration may vary according to pregnancy trimester, so why is it considered the same in this study?

line 129-133: Participants were asked to report the days per week and minutes per day in each specific activity over the past seven days. According to the International Physical Activity Inspection guidelines, PA were categorized as low-, moderate-, and vigorous-intensity PA. How was the PA calculated, and what scores correspond to low, moderate, and vigorous intensity? A more detailed explanation of the research tool is needed.

Table 1: How is household income classified? Is it the participant's subjective evaluation?

Researchers asked participants about BMI(body weight and height), regular exercise, smoking, and alcohol drinking during pre-pregnancy.  What is the specific period before pregnancy? What is the specific time before pregnancy? What is the basis for thinking that this lifestyle before pregnancy affects sleep and physical activity during pregnancy? Also, in the case of drinking, the information is very limited because participants only responded with ‘yes’ or ‘no’.

Therefore, I think that the application of the results of this study to pregnant women is very limited.

Author Response

Reviewer’s comment

Revision

Reviewer 1

The purpose of this study was to identify distinct clusters with differing combined patterns of PA and sleep quality among Korean pregnant women and examine healthy behaviors and psychological health among the identified clusters.

In this study, all pregnant women from 1st to 3rd trimesters were included. Sleep and physical activity may differ according to trimester. In this study, the difference between the three groups was not considered in the data analysis.

line 100: Do researchers view smoking and alcohol drinking as healthy behavior?

Thank you for pointing this out. We have revised the sentence as follows:

Healthy behaviors included regular exercise (30 minutes per day for three days per week), being a nonsmoker, and not drinking before pregnancy.

Line 100-101: It is not appropriate to include weight and height, planned pregnancy, and expected date due in demographic variables.

Thank you for pointing this out. Accordingly, we have revised the sentence as follows:

Demographic and pregnancy related information collected included age, education level, current employment status, household income, height, weight, planned pregnancy, and expected due date.

line 104: I suggest replacing healthy weights with normal weights.

We have replaced healthy weight with normal weight.

line 108: Need citation for PSQ

Thank you for pointing this out. We have added the appropriate reference.

We agree that the trimester of pregnancy may affect the sleep duration of women. However, asleep duration of 7–9 hours was considered to be appropriate for all pregnant women. Therefore, we used three categories in the final data analysis.

We agree with this comment. Therefore, we have added the following sentence to provide detailed information on IPAQ:

According to the International Physical Activity Inspection guidelines, the level of physical activity of participants was divided into three categories: low-, moderate-, and high PA [24]. Metabolic Equivalent Task (MET) is an indicator for energy expenditure. MET levels are defined as low intensity (3.3 METs), moderate intensity (4.0 METs), and vigorous intensity (8.0 METs). Total MET-minutes/week was computed using the following formula: low (MET x min x days) + moderate (MET x min x days) + vigorous (MET x min x days).

Yes, we asked the participants to choose one among four categories ranging from very low to very high income, corresponding to their perception of their household income.

Researchers asked participants about BMI(body weight and height), regular exercise, smoking, and alcohol drinking during pre-pregnancy.  What is the specific period before pregnancy? What is the specific time before pregnancy?

What is the basis for thinking that this lifestyle before pregnancy affects sleep and physical activity during pregnancy?

Also, in the case of drinking, the information is very limited because participants only responded with ‘yes’ or ‘no’. Therefore, I think that the application of the results of this study to pregnant women is very limited.

Thank you for pointing this out. We obtained information regarding smoking and drinking before pregnancy. Although we did not state “just before pregnancy” in the question, it could be assumed to be just before pregnancy not far before. In general, life habits during preconception are likely to continue even after pregnancy. In this study, we wanted to find modifiable factors that can intervene before pregnancy.

We agree that current drinking and the exact amount of alcohol are important to understand the effect of drinking on clusters. Therefore, we included this in the limitations section of the study as follows: As the drinking and smoking of participants was regarded only before pregnancy, the result of these effects on clusters should be understood carefully.

Reviewer 2 Report

Comments to the Author:

I thank to the editors for the opportunity to review this study, beside I would also like to congratulate the authors for the made effort in their study. The present manuscript by Lee et al., analyzed “Cluster analysis of the combined association of sleep and physical activity with healthy behavior and psychological health in pregnant women”, the authors attempted to identify distinct clusters with differing combined patterns of physical activity and sleep quality among Korean pregnant women and further examine healthy behaviors and psychological health among the identified clusters. The manuscript is interesting and covers a topic that has not been well studied. In addition, the introduction correctly shows the existing problem and provides adequate information for the reader to understand the relevance of performing such a study. Nevertheless, some important issues need to be addressed to improve the presentation of the study. 

  1. Does this study undertake in compliance with the declaration of Helsinki? Authors should add the number of ethical committee and the name of institution that approved the certificate.
  2. The exclusion and inclusion criteria should be more clearly shown.
  3. How was it controlled if smokers and drinkers did not continue to use alcohol or tobacco during pregnancy?
  4. Could the authors confirm with references whether the IPAQ test and depressive symptoms test were validated in Korean? I am asking this question because references 22 (IPAQ) and 23 (CES-D) are not references that validate the test in Korean.
  5. The authors must make an enormous effort to conduct a real discussion. The discussion section is sparse and the results of the study were not discussed with other existing studies. The authors only briefly comment on their results as in the results section. Indeed, from sentence 242 to 286, two references were just added. Authors should add more references and genuinely discuss their results and reduce the expeculative arguments.

Minor comments.

P2 L57-58: The relevance of this sentence “PA is also positively related to social relationships” in this paragraph is not understood.

P7 L244: Add reference.

Author Response

Reviewer’s comment

Revision

Reviewer 2

We agree with this. Before the study was initiated, we obtained approval from the Institutional Review Board of the women’s hospital. Information regarding IRB number is presented after the conclusion section (P9 L353-355). 

We agree with this comment. Thus, we have accordingly revised the exclusion and inclusion criteria as follows:

Inclusion criteria were women over 20 years of age, who confirmed their pregnancy by a medical doctor, and had no known physical or mental illness. Exclusion criteria included women who experienced complications during pregnancy such as gestational hypertension and diabetes mellitus.

How was it controlled if smokers and drinkers did not continue to use alcohol or tobacco during pregnancy?

We agree that the effect of smoking and alcohol during pregnancy on clusters. Therefore, we mentioned this in the study limitation. We focused more on finding factors that can be modified before pregnancy to intervene them.

Could the authors confirm with references whether the IPAQ test and depressive symptoms test were validated in Korean? I am asking this question because references 22 (IPAQ) and 23 (CES-D) are not references that validate the test in Korean.

We have added references supporting the Korean version of the scale.

(P3 L135), (P4 L151)

The authors must make an enormous effort to conduct a real discussion. The discussion section is sparse and the results of the study were not discussed with other existing studies. The authors only briefly comment on their results as in the results section. Indeed, from sentence 242 to 286, two references were just added. Authors should add more references and genuinely discuss their results and reduce the expeculative arguments.

Thank you for this pertinent suggestion. We have added more references.

This study is unique as it utilizes both sleep and physical activity of pregnant women to investigate clusters with similar patterns. Only one study with adult population is available that can be compared with our results directly. We have cited this study in the discussion. (Add references were colored in red.)

(P7 L263), (P7 L266), (P7 L270), (P7 L303)

The findings of this study also argue that cultural and contextual aspects should be considered when designing a study with pregnant women.  

Minor comments.

P2 L57-58: The relevance of this sentence “PA is also positively related to social relationships” in this paragraph is not understood.

We deleted the sentence because it did not support the paragraph.

P7 L244: Add reference.

We have added the reference. (P7 L253)

Round 2

Reviewer 1 Report

The manuscript revised according to the comments that suggested in the first review.

Author Response

Thank you for your comments.

Reviewer 2 Report

General Comments to the Author:

I commend the effort you made to revise this manuscript. However, I am not in agreement with the validation of the IPAQ test in Korea. Author’s comments reported this “We have added references supporting the Korean version of the scale”, nevertheless, the reference added by the authors does not show such validation.  (International Physical Activity Questionnaire:12-Country Reliability and Validity). The 12 cities validated by the study are: Australia, Brazil, Canada, Finland, Guatemala, Netherland, Japan, Portugal, South Africa, Sweden, US (San Diego), US (South Carolina), UK (Bristol) and UK (Cambridge). However, the study does not show any validation of this test in Korean. Therefore, it is not acceptable to conduct a study with a test that has not been validated in the language of the tested population.

Author Response

참조를 추가했습니다 (Solna, 2006). (P3 L135)

Solna, S. (2006). 국제 신체 활동 설문지. https://docs.google.com/viewer?a=v&pid=sites&srcid=ZGVmYXVsdGRvbWFpbnx0aGVpcGFxfGd4OjU1ZDdmYzVmMjk5MTg0ZDA에서 가져옴